# Peroxisomal NAD(H) Homeostasis in the Yeast *Debaryomyces hansenii* Depends on Two Redox Shuttles and the NAD^+^ Carrier, Pmp47

**DOI:** 10.3390/biom13091294

**Published:** 2023-08-24

**Authors:** Selva Turkolmez, Serhii Chornyi, Sondos Alhajouj, Lodewijk IJlst, Hans R. Waterham, Phil J. Mitchell, Ewald H. Hettema, Carlo W. T. van Roermund

**Affiliations:** 1School of Bioscience, University of Sheffield, Sheffield S10 2TN, UK; 2Laboratory Genetic Metabolic Diseases, Department of Clinical Chemistry, Amsterdam UMC, University of Amsterdam, Meibergdreef 9, 1105 AZ Amsterdam, The Netherlands; 3Amsterdam Gastroenterology Endocrinology Metabolism, Amsterdam, The Netherlands; 4Amsterdam Reproduction & Development, Amsterdam, The Netherlands

**Keywords:** redox shuttle, NAD^+^, exchanger, SLC25, solute carrier, oleaginous

## Abstract

*Debaryomyces hansenii* is considered an unconventional yeast with a strong biotechnological potential, which can produce and store high amounts of lipids. However, relatively little is known about its lipid metabolism, and genetic tools for this yeast have been limited. The aim of this study was to explore the fatty acid β-oxidation pathway in *D. hansenii*. To this end, we employed recently developed methods to generate multiple gene deletions and tag open reading frames with GFP in their chromosomal context in this yeast. We found that, similar as in other yeasts, the β-oxidation of fatty acids in *D. hansenii* was restricted to peroxisomes. We report a series of experiments in *D. hansenii* and the well-studied yeast *Saccharomyces cerevisiae* that show that the homeostasis of NAD^+^ in *D. hansenii* peroxisomes is dependent upon the peroxisomal membrane protein Pmp47 and two peroxisomal dehydrogenases, Mdh3 and Gpd1, which both export reducing equivalents produced during β-oxidation to the cytosol. Pmp47 is the first identified NAD^+^ carrier in yeast peroxisomes.

## 1. Introduction

In recent years, the halophilic yeast *Debaryomyces hansenii* has gained significant attention in biotechnology as it can grow in the presence of a high concentration of salt or ethanol, is highly resistant to fungicides and various lignocellulosic inhibitors [1,2], and has been reported to accumulate lipids to over 50% of its biomass [2,3,4]. The properties of *D. hansenii* can be utilized for medical, food, and industrial biotechnology. However, one of the limitations is that little is known about lipid metabolism in *D. hansenii*. The aim of this work was to uncover pathways and proteins involved in the fatty acid breakdown in *D. hansenii*. Genome-editing tools for *D. hansenii* have been limited. To overcome this, we recently developed a PCR-based method for gene disruption based on homologous recombination and described a genomic safe landing site for the expression of heterologous proteins [5], which allows the systematic exploration of the organization of lipid metabolism pathways in this yeast.

β-Oxidation is a cyclic process of fatty acid breakdown. Among yeast species, β-oxidation is best characterized in *Saccharomyces cerevisiae*. The enzymes of the β-oxidation pathway, acyl-CoA oxidase (Pox1 or Fox1), multifunctional 2-trans-enoyl-CoA hydratase/3-hydroxyacyl-CoA dehydrogenase (Fox2), and 3-ketoacyl-CoA thiolase (Pot1 or Fox3), in *S. cerevisiae* are compartmentalized in a single membrane-bound organelle, the peroxisome. A loss of the β-oxidation enzymes, proteins involved in substrate and product transport across the peroxisomal membrane, or proteins required for the peroxisome biogenesis results in the partial or complete inability of cells to degrade fatty acids and a failure to grow on oleate as the sole carbon source [6].

During peroxisomal β-oxidation, energy equivalents are produced as a result of the reduction of NAD^+^ to NADH. In *S. cerevisiae*, these reduction equivalents are exported out of peroxisomes via dehydrogenase-dependent redox shuttle mechanisms mediated by two peroxisomal enzymes, malate dehydrogenase (Mdh3) and glycerol-3-phosphate dehydrogenase (Gpd1). The inactivation of the shuttles results in a decrease in the NAD^+^/NADH ratio and, consequently, in a decrease in β-oxidation activity [7,8]. The β-oxidation activity in *S. cerevisiae* is modulated via the NAD(H) shuttles as the involvement of each of these shuttle mechanisms depends on the expression level of proteins involved and the growth conditions [7].

The cofactors required for the β-oxidation enzymes are imported into peroxisomes from the cytosol by specialized proteins or co-imported with fully folded proteins (reviewed in [9]). It remains unclear, however, how NAD^+^ is imported into peroxisomes in yeast or humans, but in plants, the NAD^+^/AMP anti-porter PXN was shown to import NAD^+^ into peroxisomes [10,11].

We studied the oxidation of fatty acids in *D. hansenii* and found it to be exclusively mediated by peroxisomal β-oxidation. We next focused on NAD^+^ maintenance during the β-oxidation of fatty acids. For that, a series of strains with single or multiple deletions of genes encoding peroxisomal proteins with a potential role in NAD^+^ homeostasis were generated. Using radio-labelled fatty acids, we showed that two shuttle systems and the peroxisomal membrane protein Pmp47 jointly maintain the availability of intraperoxisomal NAD^+^ required for β-oxidation. To further characterize Pmp47, we examined the effect of its heterologous overexpression on β-oxidation in *S. cerevisiae*. The combined results presented here show that Pmp47 is the first identified yeast peroxisomal NAD^+^ transporter.

## 2. Materials and Methods

### 2.1. Strains, Growth Conditions and Media

The *D. hansenii* strains used in this study are shown in Appendix A. NCYC102 and NCYC3363 were obtained from Natural Collection of Yeast Cultures, Norwich Research Park, Norwich, UK. Cells were grown in YM medium containing 3 g/L yeast extract (Formedium, Swaffham, UK), 3 g/L malt extract (Oxoid, Basingstoke, UK), 5 g/L peptone (Formedium, Swaffham, UK) and 10 g/L D-glucose (Thermo Fisher Scientific, Loughborough, UK). For solid medium, 15 g/L agar (Formedium, Swaffham, UK) was added. Where antibiotic selection was required, hygromycin B (PhytoTech Labs, UK distributor is Generon, Slough, UK)), ClonNat (Melford, Ipswich, UK) or G418 disulfate (Melford, Ipswich, UK) was added into YM medium. Amounts of 25 mg/L or 50 mg/L of hygromycin B, 1.5 mg/L or 4 mg/L ClonNat, and 150 mg/L or 350 mg/L of G418 disulphate were added for the selection of NCYC102 or NCYC3363, respectively. For G418 selection, it was important to plate out cells at a relatively low cell density to minimise background growth, and to replicate to a fresh selective plate after 24 h of incubation. The cells were grown at 25 °C, either on solid media or in liquid cultures on a shaker at 200 rpm. *S. cerevisiae* strains used in this study are listed in Appendix A.

For fluorescence microscopy, both *D. hansenii* and *S. cerevisiae* cells were grown to log phase in yeast minimal complete medium containing 5 g/L ammonium sulphate (BDH), 1.9 g/L yeast nitrogen base without ammonium sulphate and amino acids (Merck life sciences UK Ltd., Gillingham, Dorset, UK, Y1251), 20 g/L D-glucose, 10 g/L casamino acids (Formedium, Swaffham, UK), 20 mg/L uracil, 20 mg/L tryptophan and 30 mg/L leucine (Merck life sciences UK Ltd., Gillingham, Dorset, UK) or cells from glucose cultures were transferred for 6 h into oleate medium (0.125% *v*/*v* oleate, 0.2% *v*/*v* Tween-40, 1 g/L yeast extract, 5 g/L peptone, 50 mM potassium phosphate buffer (pH = 6.0)). For β-oxidation activity measurements in *D. hansenii* or *S. cerevisiae*, cells were cultured in glucose medium (glucose-grown cells) (5 g/L glucose and 1 g/L yeast extract). To induce peroxisomal proliferation (oleate-induced cells), cells were cultured for 24 h in glucose medium (5 g/L glucose and amino acids: 30 mg/L leucine, 20 mg/L uracil, or 20 mg/L tryptophan (Merck life sciences UK Ltd., Gillingham, Dorset, UK) and then transferred to and cultured overnight in oleate-based medium (25 mM potassium phosphate buffer (pH = 6.0), 1.2 g/L oleate (Fisher Scientific, Loughborough, UK), 3 g/L yeast extract (Gibco), 5 g/L peptone (Gibco), and 2 g/L Tween-80 (Merck life sciences UK Ltd., Gillingham, Dorset, UK). Lysine-deficient medium for spot assays (growth analysis) was composed of 20 g/L glucose and yeast minimal complete medium with lysine drop-out (Formedium, Swaffham, UK). Oleate medium and glycerol medium for spot assays (growth analysis) were composed of yeast minimal complete medium containing 5 g/L ammonium sulphate (BDH), 1.9 g/L yeast nitrogen base without ammonium sulphate and amino acids (Merck life sciences UK Ltd., Gillingham, Dorset, UK, Y1251), 10 g/L casamino acids (Formedium, Swaffham, UK), 20 mg/L uracil, 20 mg/L tryptophan and 30 mg/L leucine (Merck life sciences UK Ltd., Gillingham, Dorset, UK) containing either 0.125% (*v*/*v*) oleate with 0.5% (*v*/*v*) Tween-40 or 3% (*v*/*v*) glycerol.

*E. coli* DH5α or INVα cells were grown at 37 °C either as a liquid culture under shaking at 200 rpm or on solid media, using 2TY media which contain 16 g/L tryptone (Formedium, Swaffham, UK), 10 g/L yeast extract, 5 g/L sodium chloride (Fisher Scientific, Loughborough, UK) and 20 g/L agar for solid medium. Where required, ampicillin sodium salt (MP Biomedicals, Santa Anna, CA, USA) was added into the medium to a final concentration of 75 mg/L.

### 2.2. Molecular Biology Techniques

Plasmids were constructed and amplified in *E. coli* DH5α or INVα. Plasmids were purified using a QIAGEN Plasmid Miniprep kit, according to the manufacturer’s recommendations, and are described in Appendix A. The different PCR polymerases and buffers were supplied by Meridian Bioscience (formerly Bioline, Nottingham, UK) or New England Biolabs (Leiden, Netherlands). The oligonucleotides were supplied by Merck. The PCR protocols were performed as prescribed by the manufacturer and the primers used in this research are listed in Appendix A. PCR products and plasmid digests were analysed using 0.7% agarose/TAE gel electrophoresis. T4 DNA ligase and buffer were purchased from New England Biolabs (Leiden, The Netherlands). The DNA sequences of newly made plasmids were confirmed by Sanger Sequencing, which was carried out by Source Bioscience or Core Facility Genomics of Amsterdam UMC, and the results were analysed by Clustal Omega database [12] or Codoncode Aligner (version 8.0.2) by aligning both estimated and obtained plasmid sequences.

*D. hansenii* total DNA isolation was based on the method described by [13] with some minor modifications, as described in [5].

For *S. cerevisiae* plasmids, *URA3* and *LEU2* centromeric plasmids were derived from Ycplac33 and Ycplac111 [14] and they both contained the *PGK1* terminator. These ARS1/CEN4 plasmids are present at one to two copies per cell [15]. The plasmid constructs were generated either by the gap repair mechanism in yeast or by restriction digestion–ligation-based methods in *Escherichia coli*. Plasmids encoding cytosolic or peroxisomal CPT2 have been described elsewhere [16]. *DhPMP47*, *DhGPD1* and *DhMDH3* ORFs were amplified from total DNA of the strain NCYC102. The *DhMDH3* ORF contained a CTG codon, which was changed into a serine codon (TCA) that was compatible with heterologous expression in *S. cerevisiae*. This was carried out using primers VIP5183 and VIP5184 (Appendix A) and a site-directed mutagenesis kit (Agilent, Santa Clara, CA, USA, Quick change II) accordingly to the manufacturer’s instructions.

For *D. hansenii*, a clonNat resistance cassette was created to delete ORFs. The *sat-1* gene from the bacterial transposon Tn1825 encodes streptothricin acetyltransferase, which confers resistance to nourseothricin, and was amplified by PCR from pFA-SAT1 [17] to introduce a PstI site at the 5′ end of the cassette and a BamHI site at the 3′ end of the cassette using oligonucleotide VIP3286 and VIP3287, respectively. The PCR product was cloned into pBluescript KS+ (Appendix A). Additionally, pHygR and pKanR plasmids have been described in [5].

To construct a GFP-PTS1 expression plasmid that integrates into the *DhARG1* locus, the *Meyerozyma guilliermondii ACT1* promoter-with CTG codons-adapted GFP ORF [18] was amplified by PCR introducing the amino acid sequence -P-L-H-S-K-L at the c-terminus of GFP, which functions as a PTS1. This sequence was inserted into pSA5 using the Not1 and Sal1 restriction sites that were introduced by the PCR reaction, which resulted in pSA6, which contains the HygR marker and *MgACT1*pr-GFP-PTS1 that is also flanked by ~1 kb upstream and ~1 kb downstream of *DhARG1*. This plasmid was also used as a template to amplify only the region of the HygR marker and *MgACT1*pr-GFP ORF (without the PTS1) to tag proteins N-terminally in the genome. To construct a C-terminal tagging plasmid, the CTG codon-adapted GFP-PTS1 ORF from pSA6 was amplified using primers VIP4778 and VIP4779; this replaced the PTS1 in the GFP sequence with a stop codon. This fragment, together with the *SsGPD1 terminator*, was introduced into pHygR, which resulted in the plasmid pSLV38.

### 2.3. D. hansenii and S. cerevisiae Strain Construction

For the deletion of *PEX3*, *FOX2*, *MDH3* or *GPD1* in *D. hansenii*, long flanking regions between 500 bp and 1 kb in length were cloned into pHygR, pKanR or pSAT1. For the primers, see Appendix A. For the deletion of the *DhPMP47* or *DhNPY1* ORFs, 50 bp flanking regions were introduced by PCR using primers that anneal on the selectable cassette and contain a 50 nt 5′ extension identical to the flanking regions of the ORFs (see Appendix A). PCR products were transformed using electroporation as previously described [19] with some small adaptations as described in [5]. These gene deletion mutants were generated in the strain NCYC3363. All potential gene deletion mutants were screened for the presence of the gene deletion and for the absence of the WT copy before they were functionally analysed. The expression of mCherry-PTS1 was targeted to the *ARG1* locus in NCYC102 wild-type (as described in [5]) and NCYC102 *pex3::sat1*.

*S. cerevisiae* deletion strains were described previously and are listed in Appendix A. *S. cerevisiae* BJ1991 strains were used for β-oxidation measurement, BY4742 strains for acylcarnitine measurements, and BY4741 strains for a spot test analysis of growth.

The parental strains used for the construction of the described strains are named “wild-type” throughout the text.

### 2.4. Tagging in the Genome

Since no low-copy-number segregating plasmids were available, we generated integrations cassettes that allowed for GFP tagging in the *D. hansenii* genome (Appendix A). The N-terminal GFP-tagging cassette contains the constitutive heterologous *MgACT1* promoter (Appendix A) that was shown previously to direct GFP expression in *D. hansenii* [18]. The vector pSA6 was used as a template, and the region of HygR marker-*MgACT1*pr-GFP (without the PTS1 and stop codon) was amplified with PCR, using the forward primer, which introduces the last 50–60 nt upstream of the *MDH3* ORF, and the reverse primer, which introduces a (Gly-Ala)3 linker and encoded the first 50–60 nt of the *MDH3* ORF. The resulting PCR product was subsequently amplified with PCR to extend the homology arms of the tagging construct to 90–100 bp, as we noticed that the longer flanks provide a higher efficiency of targeted integration and expression. The PCR product was transformed into wild-type or *pex3Δ* cells already expressing mCherry-PTS1. The N-terminal tagging of *Dh*Npy1 was carried out in an identical way. For the C-terminal tagging of *Dh*Gpd1, the vector pSLV38, was amplified with PCR to introduce the last 60–90 bp of *GPD1* ORF without a stop codon, and the first 60–90 bp of downstream of *GPD1* ORF. The resulting PCR product was subsequently amplified by PCR to extend the homology arms. The PCR product was transformed into wild-type or *pex3Δ* cells already expressing mCherry-PTS1. This strategy was also used to tag *DhPMP47* in the genome. Colonies expressing GFP-fusion proteins were identified through microscopy and correct integration was confirmed by PCR.

### 2.5. Fluorescence Microscopy and Image Processing

Live cell imaging of *D. hansenii* and *S. cerevisiae* cells was performed as described previously [20]. Essentially, cells were imaged with an inverted motorized microscope (Axiovert 200 M; Carl Zeiss, Oberkochen, Germany) equipped with an Exfo X-cite 120 excitation light source containing band pass filters (Carl Zeiss, Oberkochen, Germany and Chroma Technology, Bellows Falls, VT, USA), a Plan-Apochromat 63 × 1.4 NA objective lens (Carl Zeiss, Oberkochen, Germany) and a digital camera (Orca ER; Hamamatsu Photonics, Shizuoka, Japan). Image acquisition was performed using Volocity software (PerkinElmer, Waltham, MA, USA). Fluorescence images were collected as 0.5 μm *z*-stacks and presented as maximum intensity projections into one plane using Openlab software (openlab 5.5.2 demo; PerkinElmer, Waltham, MA, USA) and processed further in Photoshop (Adobe). Bright-field images were collected in one plane and processed where necessary to highlight the circumference of the cells in blue and overlaid on fluorescent images.

### 2.6. Western Blot Analysis

Western blot analysis was performed as described in [20]. GFP-tagged proteins were detected using anti-GFP (mouse IgG monoclonal antibody clone 7.1 and 13.1; 1:3000; Roche, UK, 11814460001). Actin was detected with a polyclonal anti-actin serum (rat; 1:10,000; gift by Kathryn Ayscough, School of Biosciences, University of Sheffield, UK). As secondary antibodies, HRP-linked anti-mouse polyclonal (goat; 1:4000; Bio-Rad, 1706516) and anti-rat polyclonal (rabbit; 1:10,000, Merck life sciences UK Ltd., Gillingham, Dorset, UK, A5795) antibodies were used. Detection was achieved using enhanced chemiluminescence reagents (GE Healthcare, UK) and chemiluminescence imaging. A BLUeye-prestained protein ladder (10–245 kDa) (Geneflow, Lichfield, UK) was used as a molecular weight reference.

### 2.7. β-Oxidation Measurement

The β-oxidation activity of *D. hansenii* or *S. cerevisiae* cells was measured using 10 µM of [1-^14^C]-labelled octanoate (C8:0) as substrate (American Radiolabeled Chemical, St. Louis, MO, USA), as described previously [21]. The sum of the end products of β-oxidation, which includes [^14^C]-labelled CO_2_ and acid-soluble products, was taken as a measure of β-oxidation activity.

### 2.8. Mass Spectrometric Metabolite Analyses

To compare the acyl-CoA levels in different cellular compartments, we first transformed the yeast cells with human carnitine palmitoyltransferase 2 lacking the mitochondrial targeting signal and extended with (named *Hs*CPT2^PTS1^) or without (named *Hs*CPT2^cyt^) a peroxisomal targeting signal (PTS1), as described previously [16]. After culturing for at least 24 h on 0.5 g/L glucose supplemented with amino acids (20 mg/L), yeast cells were cultured overnight on oleate-based medium (2.4 mM oleate (C18:1; 99% Merck), 2 g/L Tween-80, 3 g/L yeast extract, 5 g/L bacto peptone, 25 mM potassium phosphate, and 8 mM carnitine, (pH = 6.0)). Of this cell suspension, 2 mL was spun down and the cell pellets were washed with PBS. The pellets were stored at −80 °C.

For acylcarnitine measurements, the cell pellets were taken up in 200 μL of 70% (*v*/*v*) acetonitrile and 50 µL of 1.89 µM [^2^H_3_] palmitoyl-carnitine (internal standard) and evaporated under a steam of nitrogen at 40 °C; the resulting pellet was dissolved in 100 µL methanol. Acyl-carnitine levels were determined using HPLC-tandem mass spectrometry as described previously [22].

## 3. Results

### 3.1. Fatty Acid Degradation Occurs Exclusively via Peroxisomal β-Oxidation in D. hansenii

In *S. cerevisiae*, peroxisome numbers increase when cells are shifted from a glucose- to oleate-based medium [23]. This increase in peroxisome numbers in response to growth on oleate-based medium is also observed in wild-type *D. hansenii* cells (NCYC102) expressing the artificial peroxisomal marker mCherry-PTS1 [5] (Figure 1A), suggesting an important role of peroxisomes in the utilization of fatty acids as a carbon source in this yeast. To test whether peroxisomes are essential for fatty acid breakdown in *D. hansenii*, we deleted the evolutionarily conserved *PEX3* gene, that is required for the formation of peroxisomes, using a completely heterologous selection cassette that was previously used in *Candida albicans* ([17,24]; see also Appendix A). The deletion of *PEX3* led to the mislocalisation of the peroxisomal marker mCherry-PTS1 (Figure 1A), confirming the absence of peroxisomes in *pex3*∆ cells. We cultured cells on a medium with oleate as the sole carbon source and compared this to growth on glucose- and glycerol-containing media (Figure 1B). Whereas wild-type cells grew well on the three media tested, *pex3*∆ cells failed to grow on oleate, implying an essential role of peroxisomes in fatty acid utilization. To identify proteins potentially involved in peroxisomal fatty acid β-oxidation in *D. hansenii*, we searched the *D. hansenii* CBS767 proteome (Uniprot proteome ID, UP000000599) for proteins with a peroxisomal targeting signal type 1 (PTS1) [(S/A/C/N/P/Q/E/V)-(K/R/H/Q/N/S)-(L/M/I/F) [25] motif at the C-terminus)] or type 2 (PTS2) [-(R/K)(L/V/I/Q)-X-X-(L/V/I/H/Q)-(L/S/G/A/K)-X-(H/Q)-(L/A/F) [26] motif near the N-terminus] using the Scan Prosite database [27,28] and we used reciprocal BLAST searches to identify *D. hansenii* orthologues of known β-oxidation proteins of *S. cerevisiae*, *Ustilago maydis*, or *Homo sapiens* (Appendix A). Analogous to *S. cerevisiae*, the predicted proteome of *D. hansenii* lacks typical mitochondrial β-oxidation enzymes but a full set of potential peroxisomal β-oxidation enzymes was identified (Appendix A).

Multiple potential paralogues were identified for acyl-CoA oxidase and 3-ketoacyl-CoA thiolase, and one orthologue of Fox2, responsible for the second and third reactions of the β-oxidation pathway. We generated a *FOX2* deletion mutant and cultured this overnight on a medium containing 1.2 g/L oleate and subsequently measured the degradation of [1-^14^C]-labelled octanoate (C8:0). Similar to what was observed in *S. cerevisiae* cells, oleate-induced *D. hansenii* cells displayed a much higher rate of β-oxidation than glucose-grown cells (Figure 1C,D). The loss of Fox2 resulted in the complete inability of cells to degrade the fatty acids (Figure 1C,D). In line with this, *fox2*∆ cells were able to grow on glucose and glycerol medium but not on a medium with oleate as the sole carbon source (Figure 1B). Based on these combined results, we conclude that fatty acid β-oxidation is restricted to peroxisomes in *D. hansenii*.

### 3.2. Mdh3 and Gpd1 Are Localized to Peroxisomes but Not Essential for β-Oxidation Activity in D. hansenii

We next studied the peroxisomal NAD^+^ homeostasis in *D. hansenii.* Two dehydrogenases with peroxisomal targeting signals were identified, malate dehydrogenase 3 (Mdh3) with a potential PTS1 and glycerol-3-phosphate dehydrogenase 1 (Gpd1) with a potential PTS2 (Appendix A). To confirm that Mdh3 was present in *D. hansenii* peroxisomes, we expressed GFP-Mdh3 in wild-type and *pex3Δ* cells already expressing the peroxisomal marker protein mCherry-PTS1. This showed a clear GFP punctate pattern in the wild-type cells, colocalising with mCherry-PTS1, while in the peroxisome-deficient *pex3Δ* cells, GFP-Mdh3 showed a diffuse cytosolic labelling mirroring that of mCherry-PTS1 (Figure 2A). To confirm that Gpd1 was present in *D. hansenii* peroxisomes, we expressed a Gpd1-GFP construct in the wild-type and *pex3Δ* cells expressing the peroxisomal marker protein mCherry-PTS1. Gpd1-GFP and mCherry-PTS1 colocalised in a punctuated pattern in the wild-type cells, while in the *pex3*∆ cells, both were predominantly localized in the cytosol, confirming that Gpd1 is a peroxisomal enzyme in *D. hansenii* (Figure 2B). Collectively, these results show that both Gpd1 and Mdh3 localise to peroxisomes in *D. hansenii*.

*S. cerevisiae* NAD(H) shuttles, comprising Mdh3 and Gpd1, have been shown to be essential for the β-oxidation of fatty acids [7]. To study if the same is true for Mdh3 and Gpd1 in *D. hansenii*, we generated mutants with either *MDH3* or *GPD1* or both genes deleted. Surprisingly, growth on oleate medium (Figure 2C) and β-oxidation activity were not affected in the *mdh3Δ*, *gpd1Δ*, or even *mdh3Δgpd1Δ* cells compared to the wild-type cells (Figure 1C,D). In contrast, in *S. cerevisiae*, β-oxidation was strongly impaired in the *mdh3Δ* and *mdh3Δgpd1Δ* cells compared to the wild-type cells (22% and 10% of the activity measured in the wild-type cells, respectively (Figure 1D)). These results show that Mdh3 and Gpd1 are not essential for peroxisomal β-oxidation in *D. hansenii* and suggest the existence of an independent mechanism(s) to maintain peroxisomal NAD^+^ level.

### 3.3. Pmp47, Together with Mdh3 and Gpd1, Is Essential for Normal β-Oxidation

Based on BLAST analysis, we identified an uncharacterized membrane protein with homology to the mitochondrial carrier family. Close homologues include the peroxisomal proteins Pmp47 from *Candida boidinii* [29] and PXN from *Arabidopsis thaliana* [10,11], but no close *S. cerevisiae* homologue was identified. However, in a BLAST search in the Saccharomyces Genome Database, the highest homology was detected with *S. cerevisiae* mitochondrial NAD^+^ transporter proteins YIA6 and its paralogue YEA6 [30]. To confirm the peroxisomal location of Pmp47, we tagged Pmp47 at its carboxy-terminus in the genome with GFP, grew the cells on oleate medium and observed a clear GFP membrane labelling surrounding the peroxisomal lumen stained with mCherry-PTS1 (Figure 3A). As the transcription of genes encoding proteins involved in fatty acid β-oxidation is strongly induced when yeast cells are grown on oleate media [31], we tested whether the protein levels of Pmp47 and also Gpd1 were also induced on oleate. As Pmp47 and Gpd1 are tagged at their carboxy terminus with GFP, they are still controlled by their own promoter. We found that the levels of both Pmp47-GFP and Gpd1-GFP were increased in oleate-grown cells compared to glucose-grown cells, which supports their role in fatty acid degradation (Figure 3B).

Based on its homology with *S. cerevisiae* mitochondrial NAD^+^ transporter proteins YIA6 and its paralogue YEA6, we hypothesized that Pmp47 could be an NAD^+^ transporter. To test this, we generated cells with deletions of *PMP47* alone and in combination with the genes encoding the peroxisomal dehydrogenases, *MDH3* and *GPD1*, and tested the mutants for their ability to grow on oleate medium. Whereas cells deficient of Pmp47 alone or Mdh3 and Gpd1 grew normally on oleate medium, the growth of *mdh3Δgpd1Δpmp47Δ* cells was affected (Figure 3C). Subsequently, we studied the oxidation of [1-^14^C]-octanoate in these cells. As shown in Figure 1D, the deletion of *PMP47* did not affect β-oxidation activity. However, the loss of *PMP47* together with the loss of *MDH3* and *GPD1* resulted in a severe decrease in β-oxidation activity (Figure 1D), thereby indicating that Pmp47 contributes to peroxisomal fatty acid β-oxidation, although its function can only be shown in cells in which both peroxisomal redox shuttles are deficient. These combined results strongly indicate that NAD^+^, required for β-oxidation inside the peroxisomes of *D. hansenii* cells, is supplied by genetically redundant pathways: the reoxidation of NADH via two different redox shuttles and direct import by an NAD^+^ transporter Pmp47.

### 3.4. Pmp47 Is a Peroxisomal NAD^+^ Transporter

Previously, we characterised the peroxisomal NAD^+^/AMP transporter protein PXN from *Arabidopsis thaliana* and showed that it can functionally complement *S. cerevisiae* deletion mutants affected in peroxisomal NAD^+^ metabolism [11]. As Pmp47 is homologous to *At*PXN, and since our results support a role for Pmp47 as an NAD^+^ transporter, we adopted the same approach for Pmp47.

First, we used *S. cerevisiae* mutants with a deficiency of one (*mdh3Δ* or *gpd1Δ*), or both (*mdh3Δgpd1Δ*) peroxisomal NAD(H) shuttles. Cells deficient of Mdh3 or both Mdh3 and Gpd1 have a lower intraperoxisomal NAD^+^/NADH ratio and, as a result, a decreased β-oxidation [7]. Pmp47 or Pmp47-GFP from *D. hansenii* were stably expressed in the cells. Similarly to *D. hansenii* cells, Pmp47-GFP localized to peroxisomes in *S. cerevisiae*, as evidenced by the colocalisation with the peroxisomal marker Pex11-mRFP (Figure 4A). When we measured [1-^14^C]-octanoate β-oxidation activity, we found that the heterologous expression of *Dh*Pmp47 completely rescued [1-^14^C]-octanoate β-oxidation activity in *mdh3Δ* and *mdh3Δgpd1Δ* cells (Figure 4B). In line with the β-oxidation measurements, the heterologous expression of *Dh*Pmp47 restored the ability of *mdh3Δ* and *mdh3Δgpd1Δ* cells to utilise oleate as the sole source of carbon (Figure 4C). This clearly shows that heterologously expressed *Dh*Pmp47 supplies *S. cerevisiae* peroxisomes with NAD^+^.

We previously showed that *A. thaliana* PXN’s ability to restore β-oxidation in *S. cerevisiae mdh3Δ* cells relies on the peroxisomal nudix hydrolase, Npy1, that converts NADH to AMP and NMNH [11,32]. Surprisingly, however, we found that Pmp47 expression restored β-oxidation in *mdh3Δnpy1Δ* cells (Figure 4B). Moreover, although *Dh*Npy1 is a peroxisomal protein (Appendix A), β-oxidation activity and growth on oleate medium are not affected in *D. hansenii mdh3Δgpd1Δnpy1Δ* cells, in contrast to *mdh3Δgpd1Δpmp47Δ* cells (Figure 1D and Appendix A). Taken together, our results strongly indicate that *Dh*Pmp47 does not require Npy1 activity to provide peroxisomes with NAD^+^, and thus does not need to exchange NAD^+^ for AMP or NMNH to revert the β-oxidation deficiency of the *mdh3Δ* and *mdh3Δgpd1Δ S. cerevisiae* cells.

In *S. cerevisiae mdh3Δ* cells, β-oxidation is affected at the dehydrogenation step, leading to the accumulation of an intermediate product of oleate β-oxidation, 3-hydroxy octadecenoyl-CoA (3-hydroxy-C18:1-CoA) [8]. To further strengthen our conclusion that *Dh*Pmp47 is an NAD^+^ transporter, we aimed to measure the accumulation of 3-hydroxy octadecenoyl-CoA. To measure the acyl-CoA esters present specifically in the peroxisomal lumen, we targeted the human mitochondrial protein carnitine O-palmitoyltransferase 2 (CPT2) to the peroxisomal lumen (CPT2^PTS1^) in the different *S. cerevisiae* cells (Figure 4D) [16]. CPT2 converts all acyl-CoA esters, including 3-hydroxy acyl-CoA, to carnitine esters, which can then be measured using tandem mass spectrometry. This revealed a significantly higher intraperoxisomal 3-hydroxy octadecenoyl-carnitine level in *mdh3Δ* cells than in wild-type cells when grown overnight on the oleate medium (Figure 4E). The increase in intra-peroxisomal 3-hydroxy octadecenoyl-carnitine levels was fully reverted upon the expression of Pmp47 (Figure 4E), further supporting our hypothesis that *Dh*Pmp47 supplies peroxisomes with the NAD^+^ required for the dehydrogenation step of β-oxidation.

### 3.5. Pmp47 Expression Restores Lysine Bradytrophy in S. cerevisiae mdh3Δgpd1Δ cells

In *S. cerevisiae*, the Mdh3- and Gpd1-dependent redox shuttles are not only required for fatty acid β-oxidation but also for lysine biosynthesis. The ultimate step in lysine biosynthesis, i.e., the reduction of saccharopine to lysine, is catalysed by NAD^+^-dependent saccharopine dehydrogenase and, thus, lysine biosynthesis also requires intraperoxisomal NAD^+^ (Figure 5A). *S. cerevisiae mdh3Δgpd1Δ* cells are, therefore, lysine bradytroph (Figure 5B). Our hypothesis that Pmp47 imports NAD^+^ into peroxisomes predicts that the lysine bradytrophy of *S. cerevisiae mdh3Δgpd1Δ* cells should be reversed upon the expression of *Dh*Pmp47. Indeed, the slow-growth phenotype on the lysine-deficient medium was restored by the heterologous expression of *Dh*Pmp47 or the redox shuttle enzymes *Dh*Mdh3 or *Dh*Gpd1 (Figure 5B). This finding further supports our model that *Dh*Pmp47 can supply peroxisomes with NAD^+^, as NAD^+^ is the only known metabolite required for both fatty acid β-oxidation and the activity of saccharopine dehydrogenase.

## 4. Discussion

Whereas in *S. cerevisiae*, all the enzymes required for the β-oxidation of fatty acids and some of the transporters have been identified [6,33], it has remained unclear how yeast peroxisomes are supplied with some essential cofactors [9]. For instance, an NAD^+^ transporter in *S. cerevisiae* has not been identified. When we searched for the orthologues of known peroxisomal proteins in *D. hansenii*, it became apparent that, similarly to in *S. cerevisiae*, fatty acid β-oxidation takes place exclusively in peroxisomes. During fatty acid β-oxidation, NAD^+^ is reduced to NADH. In *S. cerevisiae* cells, the peroxisomal NADH is reoxidised to NAD^+^ by malate dehydrogenase (Mdh3) or glycerol-3-phosphate dehydrogenase (Gpd1), and reduction equivalents in the form of malate or glycerol-3-phosphate are transferred to the cytosol. We found that the orthologues of Mdh3 and Gpd1 in *D. hansenii* were also present in peroxisomes but, in contrast to *S. cerevisiae*, they are not essential for β-oxidation. Interestingly, in human cells, similarly to *D. hansenii* cells, two redox shuttle systems regulate peroxisomal NAD^+^/NADH ratio, but they are not essential for β-oxidation [34].

In this study, we showed that the *D. hansenii* protein Pmp47 is a peroxisomal NAD^+^ transporter and is able to (1) restore the β-oxidation activity, (2) decrease an accumulation of the intermediate product of oleate β-oxidation-3-hydroxy octadecenoyl-CoA, and (3) reverse the lysine bradytrophy caused by the dysfunction of the NAD^+^ reoxidation machinery in *S. cerevisiae*. Moreover, the deletion of *PMP47* together with *MDH3* and *GPD1* in *D. hansenii* leads to a decrease in β-oxidation activity. Pmp47 is an integral membrane protein [29] which, based on sequence homology, is a member of the mitochondrial solute carrier protein family. An earlier study showed that the heterologous expression of PMP47 from *Candida boidinii* resulted in peroxisomal localization in *S. cerevisiae* [29,35] and *Hansenula polymorpha* [36]. Apart from the peroxisomal localization of Pmp47 and its homologues in different organisms, there is evidence that these membrane proteins have a role in lipid breakdown in *C. boidinii* [35] as well as in pumpkin [37] and *A. thaliana* [10].

Previously, we showed using *S. cerevisiae* cells that *At*PXNp (the *A. thaliana* homologue of Pmp47) has a role in β-oxidation as an NAD^+^/AMP or NAD^+^/NMNH carrier [11]. Furthermore, we showed that *At*PXNp expression in *S. cerevisiae* can rescue the octanoate β-oxidation in *mdh3Δ* cells only in the presence of the peroxisomal nudix hydrolase Npy1, which hydrolyses NADH into AMP and NMNH, suggesting that *At*PXN exchanges NAD^+^ with AMP or NMNH and not with NADH. The expression of *Dh*Pmp47, however, fully rescued the octanoate β-oxidation activity in *mdh3Δ* as well as in *mdh3/npy1Δ* cells, indicating that *Dh*Pmp47 is not dependent on the activity of intraperoxisomal nudix hydrolase Npy1 and suggesting that Pmp47 may exchange NAD^+^ for NADH.

A major difference in peroxisomal NAD(H) homeostasis between *D. hansenii* and *S. cerevisiae* also concerns lysine metabolism. NAD^+^-dependent saccharopine dehydrogenase (Lys1) is required for lysine biosynthesis and this enzyme is localized inside peroxisomes in *S. cerevisiae* [7,38]. However, in *D. hansenii*, the orthologue of Lys1 does not have a predicted PTS1 or PTS2 signal and therefore is most probably localized in the cytosol. In accordance with this and in contrast to *S. cerevisiae mdh3Δ* and *mdh3Δgpd1Δ* cells [7], *D. hansenii mdh3Δgpd1Δpmp47Δ* cells do not show lysine bradytrophy.

Traditionally, *S. cerevisiae* is used as a model organism to study peroxisomal fatty acid oxidation and associated human peroxisomal diseases [6]. However, BLAST analysis revealed protein orthologues of human β-oxidation-related proteins in *D. hansenii* that have not been identified in *S. cerevisiae*. In this respect, *D. hansenii* is similar to the *U. maydis*, which was suggested as a good model organism for studying human peroxisome biology [39]. However, an advantage of *D. hansenii* over *U. maydis* with respect to studying peroxisomal metabolism is that β-oxidation is restricted to peroxisomes and no redundant mitochondrial β-oxidation system is present, as has been reported for *U. maydis* [40]. Consequently, *D. hansenii* is an attractive model organism to investigate peroxisomal processes that cannot be studied in *S. cerevisiae* or *U. maydis*.

## 5. Conclusions

Fatty acid β-oxidation occurs inside peroxisomes in *D. hansenii*. Collectively, our results imply that NAD^+^, required for β-oxidation, is regenerated from its reduced form inside peroxisomes via two different redox shuttle systems and is imported from the cytosol by the NAD^+^ carrier Pmp47.

## Figures and Tables

**Figure 1 biomolecules-13-01294-f001:**
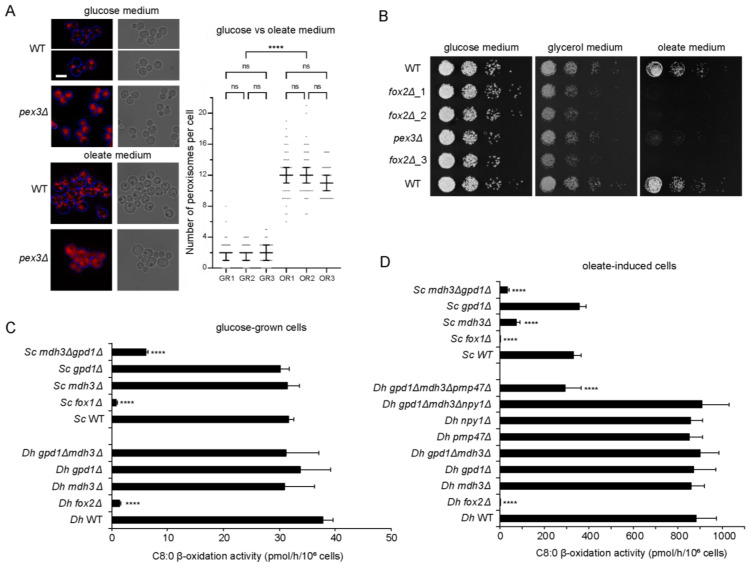
*D. hansenii* peroxisomes are required for the utilisation of fatty acids as a carbon source via β-oxidation. (**A**) Peroxisome number increased when *D. hansenii* cells were grown on oleate-based medium. Representative epifluorescence microscopy images of wild-type and *pex3Δ* cells expressing mCherry-PTS1 and grown on glucose medium to exponential growth phase or grown on oleate medium for 6 h. Red channel images are merged *z*-stacks. Peroxisome numbers were quantified for >50 budding wild-type cells per replica (n = 3, individual replicas are labelled as GR (glucose medium replica) and OR (oleate medium replica)). Bars in the graph indicate the median with 95% confidence intervals for mean. Statistical significance analysis was performed using Kruskal–Wallis test. Scale bar: 5 μm. (**B**) Growth analysis of wild type, *fox2Δ* (three independent strains) and *pex3Δ D. hansenii* cells, on media containing glucose, glycerol or oleate as sole carbon sources. (**C**,**D**) β-oxidation activity was higher in oleate-induced cells compared to glucose-grown cells, and loss of Fox2 affected β-oxidation activity in *D. hansenii* cells. NAD^+^ availability in *D. hansenii* was not solely dependent upon the Mdh3 and Gpd1 redox shuttles but also on Pmp47. The presented data relate to Section 3.1, Section 3.2 and Section 3.3 of this manuscript. β-oxidation activity in wild-type and mutant *S. cerevisiae* and *D. hansenii* cells, grown on (**C**) glucose-based medium or (**D**) oleate-based medium (oleate-induced cells), were measured using [1-^14^C]-labelled octanoate (C8:0) as substrate. The results are shown as mean ± standard deviation (n = 4). One-way ANOVA with Dunnett’s multiple comparisons test was performed to determine if differences were significant when compared with the corresponding wild-type cells. ns, not significant; **** *p* < 0.0001.

**Figure 2 biomolecules-13-01294-f002:**
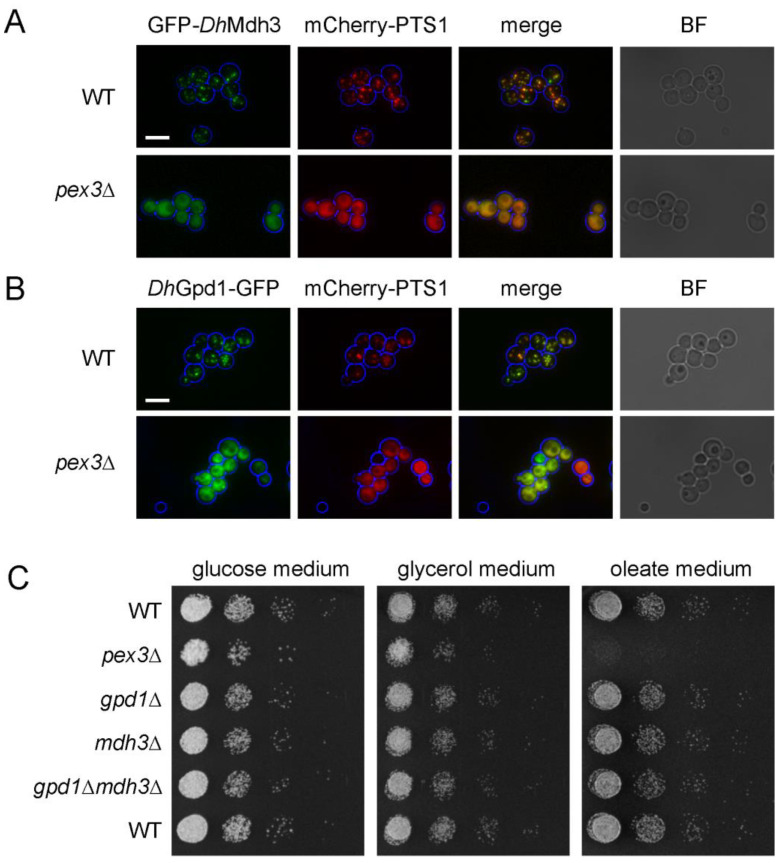
The malate and glycerol-3-phosphate dehydrogenase redox shuttles are not essential for β-oxidation in *D. hansenii*. (**A**,**B**) Representative epifluorescence microscopy images of wild-type and *pex3Δ* cells expressing mCherry-PTS1 and either (**A**) GFP-Mdh3 or (**B**) Gpd1-GFP expressed from their chromosomal loci. Cells were grown on oleate medium for 6 h. Red and green channel images are merged *z*-stacks. Merge is an overlay of the red and green merged *z*-stacks. Bright-field (BF) images were collected in one plane and were processed to highlight the cell circumference in blue in other panels. Scale bar: 5 μm. (**C**) Growth analysis of wild-type and mutant *D. hansenii* cells on media containing glucose, glycerol or oleate as sole carbon sources.

**Figure 3 biomolecules-13-01294-f003:**
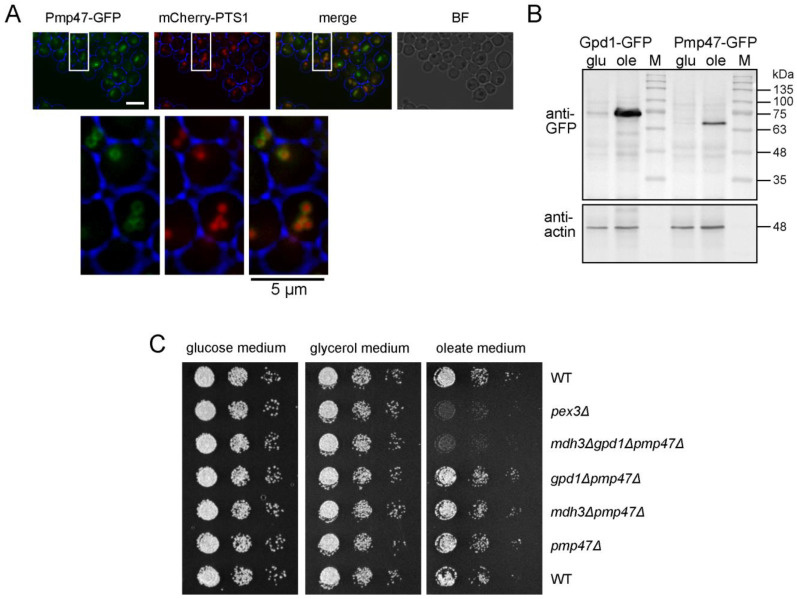
Pmp47 is a peroxisomal protein that contributes to fatty acid β-oxidation in *D. hansenii*. (**A**) Representative epifluorescence microscopy images of wild-type *D. hansenii* cells expressing mCherry-PTS1 and Pmp47-GFP expressed from its chromosomal locus. Cells were grown on oleate medium for 6 h. Red and green channel images are merged *z*-stacks. Merge is an overlay of the red and green merged *z*-stacks. Bright-field (BF) images were collected in one plane and are processed to highlight the cell circumference in blue in other panels. The white rectangles are enlarged below to show the ring-like GFP signal of Pmp47-GFP, which surrounds the peroxisomal lumen stained with mCherry-PTS1. Scale bar: 5 μm. (**B**) Western blot analysis of lysates of wild-type *D. hansenii* cells expressing Gpd1-GFP or Pmp47-GFP under control of their endogenous promoter grown on glucose (glu) or oleate (ole) media using anti-GFP and anti-actin (loading control). Protein ladder (M) was used as size standard. (**C**) Growth analysis of wild-type and mutant *D. hansenii* cells on media containing glucose, glycerol or oleate as sole carbon sources.

**Figure 4 biomolecules-13-01294-f004:**
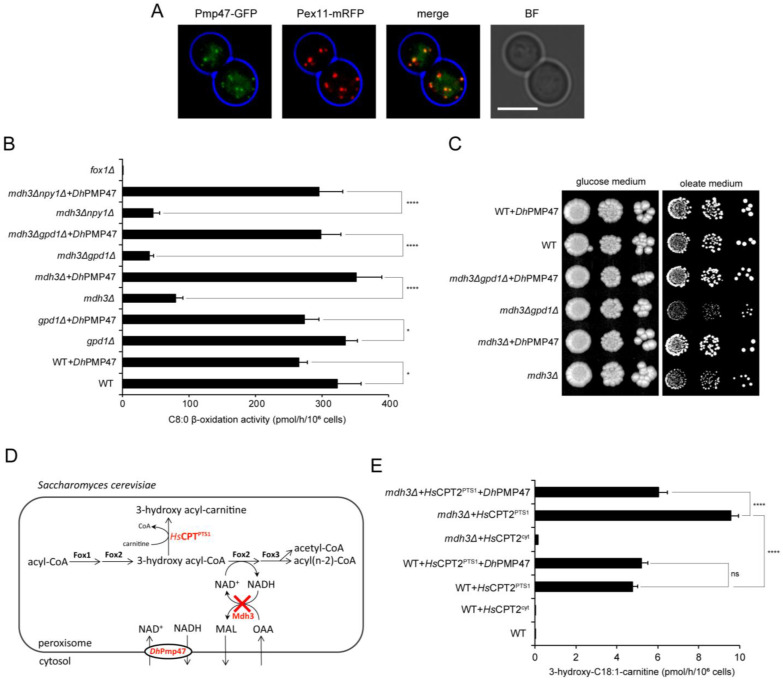
*Dh*Pmp47 is a peroxisomal NAD^+^ carrier involved in the maintenance of peroxisomal NAD^+^ required for β-oxidation. (**A**) *Dh*Pmp47-GFP was localised to the peroxisomal membrane upon expression in *S. cerevisiae* cells. Representative epifluorescence microscopy images of wild-type *S. cerevisiae* cells expressing the peroxisomal membrane marker, Pex11-mRFP, and *Dh*Pmp47 fused with GFP (*Dh*Pmp47-GFP). Red and green channel images are merged *z*-stacks. Merge is an overlay of the red and green merged *z*-stacks. Bright-field (BF) images were collected in one plane and were processed to highlight the cell circumference in blue in other panels. Scale bar: 5 μm. (**B**,**C**,**E**) *Dh*Pmp47 was expressed in wild-type or mutant *S. cerevisiae* cells behind an oleate-inducible *CTA1* promoter. (**B**) Heterologous expression of *Dh*Pmp47 rescued the β-oxidation activity of *mdh3Δ*, *mdh3Δgpd1Δ and mdh3Δnpy1Δ S. cerevisiae* cells. *S. cerevisiae* cells were oleate-induced (grown overnight on oleate medium), and β-oxidation rates were measured using [1-^14^C]-labelled octanoate as substrate. The results are shown as mean ± standard deviation (n = 4). One-way ANOVA with Tukey’s multiple comparisons test was performed to determine if differences were significant. Cells transformed with empty plasmid and *fox1Δ* cells, deficient of the first enzyme of the β-oxidation pathway, were used as controls. (**C**) Heterologous expression of *Dh*Pmp47 rescued the ability of *mdh3Δ* and *mdh3Δgpd1Δ S. cerevisiae* cells to utilise oleate. Growth analysis of the indicated *S. cerevisiae* strains either expressing *Dh*Pmp47 or transformed with an empty plasmid on media containing glucose or oleate as sole carbon source. (**D**) Schematic representation of the peroxisomal steps of fatty acid β-oxidation and main redox shuttle in *S. cerevisiae*. Deletion of peroxisomal malate dehydrogenase (Mdh3), red cross, that is part of the peroxisomal malate redox shuttle, results in accumulation of the β-oxidation intermediate 3-hydroxy acyl-CoA. To estimate the level of 3-hydroxy acyl-CoA produced in peroxisomes, acyl-CoA esters were converted to the more stable carnitine esters by heterologous expression of human carnitine O-palmitoyltransferase 2 (*Hs*CPT2) that was artificially targeted to peroxisomes by deletion of its mitochondrial targeting signal and addition of a peroxisomal targeting signal type 1. Heterologous expression of *Dh*Pmp47 provides an alternative mechanism to supply *S. cerevisiae* peroxisomes with NAD^+^. (**E**) Heterologous expression of *Dh*Pmp47 reduced 3-hydroxy octadecenoyl-carnitine levels in oleate-induced (grown overnight on oleate medium) *mdh3Δ* cells. *Dh*Pmp47 and peroxisome-targeted carnitine O-palmitoyltransferase 2 (*Hs*CPT2^PTS1^) were expressed in wild-type and *mdh3Δ S. cerevisiae* cells, as indicated. Cells expressing HsCPT2 lacking a PTS1 that localizes to the cytosol (*Hs*CPT2^cyt^) and cells transformed with empty plasmid were used as negative controls. Accumulation of the 3-hydroxy octadecenoyl-carnitine was measured using HPLC-tandem MS. The results are shown as mean ± standard deviation of 3 technical replicates. One-way ANOVA with Tukey’s multiple comparisons test was performed to determine if differences are significant. ns, not significant; * *p* = 0.01 to 0.05; **** *p* < 0.0001.

**Figure 5 biomolecules-13-01294-f005:**
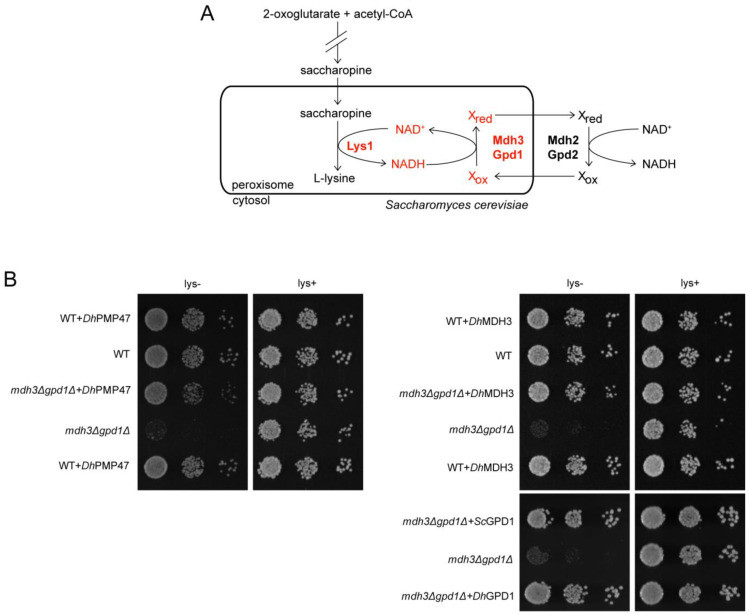
Heterologous expression of *Dh*Pmp47 supplies *S. cerevisiae* peroxisomes with NAD^+^ required for lysine biosynthesis. (**A**) Schematic representation of the peroxisomal step of lysine biosynthesis pathway in *S. cerevisiae*. A deletion of Mdh3 and Gpd1 enzymes, that are part of the peroxisomal malate and glycerol-3-phosphate redox shuttles, results in a deficiency in lysine biosynthesis. (**B**) Lysine bradytrophy (slow-growth phenotype on lysine-free medium) was repressed by heterologous expression of *Dh*Pmp47, *Dh*Mdh3 and *Dh*Gpd1 in *S. cerevisiae mdh3Δgpd1Δ* cells. Growth analysis on glucose medium in the absence or presence of lysine. Wild-type and *mdh3Δgpd1Δ* cells were transformed with plasmids encoding *Dh*Pmp47, *Dh*Mdh3, *Dh*Gpd1-GFP, *Sc*Gpd1-GFP or empty plasmid.

## Data Availability

Data are contained within the article or Appendix A.

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
