# Peer review of "Peroxisomal NAD(H) Homeostasis in the Yeast Debaryomyces hansenii Depends on Two Redox Shuttles and the NAD+ Carrier, Pmp47"

_biomolecules, 2023, doi:10.3390/biom13091294_

Round 1

Reviewer 1 Report

Dear Editor, 

The paper proposed by Turkolmez and colab. bring into the light the yeast Debaryomyces hansenii, recognized as an atypical yeast with substantial biotechnological potential and adeptness in lipid production and storage, which still harbors limited insights into its lipid metabolism. The investigation's objective was to delve into the fatty acid β-oxidation pathway within D. hansenii. In pursuit of this goal, the researchers harnessed recently developed techniques for executing multiple gene deletions and endogenous genetic locus tagging in this yeast. Their findings concur with the pattern observed in other yeasts, as the β-oxidation of fatty acids in D. hansenii have placed exclusively within peroxisomes. The paper emphasizes a series of experiments performed in D. hansenii and in the extensively studied yeast Saccharomyces cerevisiae. These experiments underscore that the equilibrium of NAD+ within D. hansenii peroxisomes hinges on the involvement of the peroxisomal membrane protein Pmp47 and two peroxisomal dehydrogenases, Mdh3 and Gpd1. Both Mdh3 and Gpd1 transport generated reducing equivalents from β-oxidation to the cytosol. In this investigation, they demonstrated that the protein Pmp47 from D. hansenii functions as a peroxisomal NAD+ transporter. This protein has the capability to reinstate β-oxidation activity, reduce the accumulation of an intermediary product of oleate β-oxidation called 3-hydroxy octadecenoyl-CoA, and reverse the lysine bradytrophy resulting from impaired NAD+ reoxidation machinery in S. cerevisiae. Furthermore, the simultaneous deletion of PMP47, MDH3, and GPD1 in D. hansenii leads to a decline in β-oxidation activity. Pmp47 is an integral membrane protein and, based on sequence similarity, is a member of the mitochondrial solute carrier protein family.

The work is well-presented and structured. The methodology, results and discussion section are very well documented.

I proposed the publication of the paper in this present form.

Author Response

Thank you for your positive review of our work.

Reviewer 2 Report

The manuscript by Turkolmez and co-workers (Peroxisomal NAD(H) homeostasis in the yeast Debaryomyces hansenii depends on two redox shuttles and the NAD+ carrier, PMP47) describes interesting results, I think within the scope of Biomolecules. However, there are some issues that need to be addressed before it can be considered for publication:

-L82-83 the concentration of the antibiotics used needs to be described.

-In the description of the medium, sometimes they refer to “oleate medium”, “oleic acid” (L91), it should be consistent through the manuscript.

-In some instances “yeast extract” is from Formedium (L79-80), in others is from Gibco (L98)… is there a reason for this?

-L133-134, how the CTG codon was changed to TCA? details are required.

-In several instances “….” are used in the text (L144-145, L148, L178-179)… why?

-There is something strange/nonsense with Figure 1D…. how could the authors measure β‑oxidation capacity in the Dh-fox2Δ strain in “oleate-grown cells”, if this Dh-fox2Δ strain does not grow in oleate medium (Fig. 1B)!. In “glucose-grown cells” (Fig. 1C) is OK, but in Fig. 1D? (see also L266-268, indeed, not surprisingly the result reported is 0, but is not clear). The same happens with the Dh-mdh3Δgpd1Δpmp47Δ cells that do not grow on oleate (Fig. 3C), but the authors report β‑oxidation capacity in the Dh-mdh3Δgpd1Δpmp47Δ strain in “oleate-grown cells” (Fig. 1D, no data is presented for this strain in “glucose-grown cells”). The authors need to better explain/justify this results/data……

-Fig. 1A, the first 2 panels (wt) are split by a line…. is there a reason for this?

-Figure 5B, the growth of WT-DhPMP47 cells is reported twice (upper and lower spots in the left panel), the same happens with WT-DhMDH3 cells (upper and lower spots in the upper right panel). Why? they are different “WT” cells? This needs to be better explained or corrected.

Author Response

-L82-83 the concentration of the antibiotics used needs to be described.

Thank you for pointing this out. We have specified the concentrations.

-In the description of the medium, sometimes they refer to “oleate medium”, “oleic acid” (L91), it should be consistent through the manuscript.

We have changed “oleic acid” to “oleate” for consistency.

-In some instances “yeast extract” is from Formedium (L79-80), in others is from Gibco (L98)… is there a reason for this?

The experiments described in this article were performed at two different locations, the University of Sheffield (growth analysis) and Amsterdam UMC (beta-oxidation measurements). The suppliers of yeast extract at both locations were different, but we believe that any of these yeast extracts can be used to culture D. hansenii or S. cerevisiae cells with similar outcome.

-L133-134, how the CTG codon was changed to TCA? details are required.

After the open reading frame encoding the DhMDH3 was subcloned into pEW324 plasmid, the CTG codon was changed using primers VIP5183 and VIP5184 (Supplemental Table 4) and site-directed mutagenesis kit (SDM: Agilent (USA) Quick change II) accordingly to manufactures instructions.

-In several instances “….” are used in the text (L144-145, L148, L178-179)… why?

Thank you for pointing this out. That was a typo, we have now removed “” symbols from the text.

-There is something strange/nonsense with Figure 1D…. how could the authors measure β‑oxidation capacity in the Dh-fox2Δ strain in “oleate-grown cells”, if this Dh-fox2Δ strain does not grow in oleate medium (Fig. 1B)!. In “glucose-grown cells” (Fig. 1C) is OK, but in Fig. 1D? (see also L266-268, indeed, not surprisingly the result reported is 0, but is not clear). The same happens with the Dh-mdh3Δgpd1Δpmp47Δ cells that do not grow on oleate (Fig. 3C), but the authors report β‑oxidation capacity in the Dh-mdh3Δgpd1Δpmp47Δ strain in “oleate-grown cells” (Fig. 1D, no data is presented for this strain in “glucose-grown cells”). The authors need to better explain/justify this results/data……

Sorry for the confusion, and thank you for pointing this out. Prior to the beta-oxidation measurements, we cultured cells for 24 h in a glucose medium and then transferred cells to an oleate medium and incubated them overnight (L97-103). Although some of the strains cannot grow on oleate as a sole source of carbon, the oleate medium used for this assay also contains yeast extract and after the overnight incubation (~17h) these cells are alive and actually dividing (growing), presumably at least until they exhaust the accumulated glycogen or trehalose inside the cells and nutrients present in the yeast extract. After the overnight incubation in an oleate medium, the same number of cells (1 OD600nm) was used in the beta-oxidation assay. This method is long established for beta-oxidation assays in yeast and has been used by us since the 1990s. We provide a detailed protocol of the beta-oxidation measurements in oleate-induced (“oleate-grown”) cells in Reference 21 (van Roermund and Hettema, 2023).

To make it less confusing, we have now changed “oleate-grown cells” to “oleate-induced cells” in Fig. 1,3 and throughout the manuscript.

-Fig. 1A, the first 2 panels (wt) are split by a line…. is there a reason for this?

We have combined two independent images to present more cells in the figure.

-Figure 5B, the growth of WT-DhPMP47 cells is reported twice (upper and lower spots in the left panel), the same happens with WT-DhMDH3 cells (upper and lower spots in the upper right panel). Why? they are different “WT” cells? This needs to be better explained or corrected.

These are the same strains. We frequently put WT strain (or other positive control) on both sides to show equal growth on the whole plate.

Reviewer 3 Report

The article Peroxisomal NAD(H) homeostasis in the yeast Debaryomyces hansenii depends on two redox shuttles and the NAD+ carrier, PMP47 demonstrates that fatty acid b-oxidation in halophilic yeast occurs in peroxisomes, just as it does in S. cerevisiae; however, they report differences in peroxisomal proteins involved in NAD homeostasis that have no orthologs in budding's yeast and are more reminiscent of b-oxidation in humans.

My only concern is that Figure 1 needs to be better described in the text and figure caption, as they report double and triple mutants described much later in the text. I suggest separating the figure and putting those mutants in the appropriate part of the paper.

Author Response

We believe that the beta-oxidation results measured in mutant cells should be reported together with positive and negative controls (wild type and fox1/2Δ cells, respectively). Therefore, the suggested separation of the data would require us to replicate the wild type and fox1/2Δ beta-oxidation activity data in every graph and thus present the same data more than once in the manuscript, and we would prefer to avoid this. We hope the reviewer can agree with this point of view.

However, based on the reviewers suggestion, we have indicated in the figure's legend that the presented data relates to the sections 3.1, 3.2 and 3.3 of this manuscript.